# The Implications of Health Disparities: A COVID-19 Risk Assessment of the Hispanic Community in El Paso

**DOI:** 10.3390/ijerph20020975

**Published:** 2023-01-05

**Authors:** Carina Cione, Emma Vetter, Deziree Jackson, Sarah McCarthy, Ernesto Castañeda

**Affiliations:** 1Center for Latin American and Latino Studies, American University, Washington, DC 20016, USA; 2Department of Sociology & Anthropology, George Mason University, Fairfax, VA 22030, USA; 3Department of Sociology, Indiana University Bloomington, Bloomington, IN 47405, USA; 4Department of Sociology, State University of New York at Albany, Albany, NY 12222, USA; 5Department of Sociology, Center for Latin American and Latino Studies, Immigration Lab, Center for Health, Risk, and Society, American University, Washington, DC 20016, USA

**Keywords:** health equity, COVID pandemic, Hispanic health, immigrant, minority health

## Abstract

Since the outbreak of the COVID-19 pandemic in the United States, Latinos have suffered from disproportionately high rates of hospitalization and death related to the virus. Health disparities based on race and ethnicity are directly associated with heightened mortality and burden of illness and act as underlying causes for the staggering impacts of COVID-19 in Latin communities in the United States. This is especially true in the city of El Paso, Texas, where over 82% of the population is Hispanic. To ascertain the level of danger that COVID-19 poses in El Paso, we constructed a point-in-time risk assessment of its Latin population and assessed a Latin individual’s likelihood of hospitalization or death related to COVID-19 by comparing relevant health profiles with high-risk co-morbidities that the Centers for Disease Control (CDC) identified in 2020. Data for this risk assessment come from 1152 surveys conducted in El Paso. The assessment included comprehensive demographic, socioeconomic, and health data to analyze disparities across Hispanic sub-populations in the city. Results revealed that around 49.3% of Hispanics in the study had been previously diagnosed with a high-risk co-morbidity and therefore have an increased likelihood of hospitalization or death related to COVID-19. Additional factors that led to increased risk included low income, homelessness, lack of U.S. citizenship, and being insured. The findings from this study additionally demonstrate that structural inequality in the U.S. must be addressed, and preventive measures must be taken at local and state levels to decrease the mortality of pandemics. Baseline population health data can help with both of these goals.

## 1. Introduction

Existing health disparities have been exacerbated by COVID-19 [1,2]. Long before the pandemic, race and ethnicity were proven to be associated with life expectancy, mortality, and the burden of illness in the United States [3]. Communities of color consistently lack access to care, proper treatment, care provider diversity, and the resources that exist in high-income and White-populated areas of the country [3]. These disparities cause and prolong poor health in minority communities and result in higher-risk individuals with multiple risk factors beyond old age [2]. This is echoed by public health professionals who observe Black and Hispanic individuals die at higher rates because of an increased likelihood of underlying health conditions and structural barriers to healthcare [1]. Gaps in healthcare increase the vulnerability of groups of people, and their existence springs from discrimination, inequality, and structural racism. 

Institutions that place lesser value on the lives of Black and Hispanic people not only turn a blind eye to health disparities but restrict minorities to riskier jobs where they work in dangerous conditions. Research in the United States, Canada, and Europe shows that immigrants and other minority communities bear higher rates of work-related accidents, illnesses, and deaths because of their over-representation in high-risk occupations [4]. Because of these factors and being less likely to work remotely, they came into contact with COVID-19 before other racial groups and experienced its impacts earlier in the pandemic. 

This article examines COVID-19 risk factors in the context of the Hispanic-majority border city of El Paso, Texas. To ascertain the level of danger that COVID-19 poses for the Hispanic community in El Paso, we constructed a point-in-time risk assessment of its Latin population. We assessed a Latin individual’s likelihood of hospitalization or death related to COVID-19 by comparing relevant health profiles with high-risk co-morbidities that the Centers for Disease Control (CDC) identified in 2020 [5]. El Paso is an important research site on health disparities, as proven by prior research documenting that the border city’s Hispanic population faces compounded structural inequalities, high risk of morality, and poor health outcomes [6,7].

Our study contributes to understanding how racial, ethnic, socioeconomic, and other intersectional experiences are predictive of health outcomes within regional and historical contexts. First, we provide a brief overview of how the COVID-19 pandemic has unfolded in El Paso and its “sister city” of Ciudad Juárez, Mexico. Then, we describe the methodology employed in gathering comprehensive survey data on Hispanics living in El Paso and compiling our risk assessment. The subsequent section analyzes significant demographic, socioeconomic, and health factors that encompass essential determinants of health and risk associated with COVID-19. These measures allowed us to predict Hispanic El Pasoans’ high risks of hospitalization and death related to COVID-19 before the pandemic struck El Paso, and thus demonstrate the foreseen impact of COVID-19 on the Latin community in the U.S. [8,9,10,11]. In doing so, we also acknowledge and explore structural forms of discrimination and violence against communities of color, particularly Latinos, that contribute to health disparities.

In the early months of 2020, cities and entire regions around the world declared public health emergencies while COVID-19, also known as coronavirus, stole millions of lives and disabled many more [12]. Its arrival in the United States was swift. Former President Donald Trump, who dissolved the White House pandemic response team in 2018, assumed a perilously disinterested approach to information about the virus spreading in Asia. Reuters estimated a COVID-19 death count of at least 69,457 in the United States on 5 May 2020 [13]. This number continued to rise, reaching 92,038 on 20 May. This marked the United States as the country with the highest COVID-19 death toll in the world [13]. The number of lives lost to COVID-19 exceeded that of the United Kingdom, then the country with the second-highest mortality rate, by over 57,000 [13]. In 2020, the total estimated number of cases in the U.S. was also the highest across the globe at 1,189,198, outnumbering China by more than 1 million cases, despite China’s population being approximately four times larger [13]. However, not all geographical areas were impacted at the same time. For example, the pandemic reached New York City many months before El Paso, Texas.

The great loss that the U.S. experienced in 2020 was also reflected in local cities’ rates of hospitalization and death related to COVID-19, which tracked demographic data and allowed researchers to glimpse the differences in rates across ethno-racial groups. In New York, the Bureau of Communicable Disease Surveillance System reported that, as of 16 April 2020, the death rates for Blacks were 92.3 per 100,000 people, and 74.3 per 100,000 for Latinos [14]. In comparison, the White and Asian death rates were 45.2 and 34.5 per 100,000 people, respectively [14]. Just two months later, the mortality and infection rates skyrocketed. At least 5322 Latinos had died from COVID-19, the majority of whom were ages 65 and above [15]. This was the highest number of lab-confirmed deaths of a racial group in New York City, exceeding that of the White population by almost 1000 [15]. Even so, these numbers were understood to be low estimates since the data cover only lab-confirmed cases, effectively excluding asymptomatic and non-lab-confirmed cases.

As the COVID-19 pandemic continued to unfold and healthcare workers fought to save lives, a familiar pattern arose that concerned researchers and advocates: Black and Hispanic individuals were not only suffering from more infections but also dying at incredibly higher rates than White and Asian people [14,15].

## 2. El Paso, Texas 

El Paso is city in the western corner of Texas, bordering Las Cruces, New Mexico, and directly to the south, Ciudad Juarez, Mexico. El Paso City’s population was around 649,000 in 2010 and 679,000 in 2020 and El Paso County has over 800,000 residents. El Paso is around 83% Hispanic. El Paso is a majority Latin city, with over 24% of the population being foreign-born [16]. It also includes Hispanics who have been in the area for decades and many generations, thus making it a great place to study the impact of health disparities within the same ethno-racial category. Poverty is directly related to health; this is visible in the U.S.–Mexico border [17]. 

El Paso experienced mounting pressure as time passed in the first months of the pandemic. Sixty-five people were hospitalized in El Paso in the first week of May 2020, and 17 were put on ventilators [18]. Local public health officials worried El Paso would suffer from limited resources, as the county only had 285 licensed ICU hospital beds [18]. Unfortunately, the City Director of Public Health, Robert Resendes, resigned on 4 May, and his replacement had not been selected at the time of his departure. The city insisted that his resignation would not negatively impact preventative action since the Office of Emergency Management handles public health crises, but the community continued to buzz with concern given that Texas was one of the top ten most infected states, with over 32,954 cases in 2020 [19,20].

Ciudad Juárez, which sits on the Mexican side of the border right next to El Paso, was also grappling with an upward trajectory of COVID-19 cases. El Paso and Ciudad Juárez are sister cities that are economically and socially intertwined. The first case of COVID-19 in the Mexican state of Chihuahua was confirmed in Ciudad Juárez on 17 March 2020, not long after cities in the U.S. began issuing public health mandates, mandatory quarantines, and other lockdown procedures [21]. Since then, the official response paralleled somewhat that of the U.S. because of the lax approach that President Andrés Manuel López Obrador took to prepare for outbreaks. He refused to close the Mexican border to visitors, instead allowing Americans and foreign tourists into the country, even though the U.S. imposed major entry restrictions at its legal entry points [22].

The border cities of Tijuana and Juárez, where Mexican and U.S. nationals travel to and from every day, and the popular destination of Cancún, had the three highest rates of documented COVID-19 cases in Mexico between January and May 2020 [23]. Things grew so dire that Armando Cabada, mayor of Ciudad Juárez, wrote to the foreign affairs secretary to ask that they block Americans from filtering into the city and spreading the coronavirus [22]. Another significant factor that the Mexican government had to consider was the Migrant Protection Protocols, or “Remain-in-Mexico”, U.S. program. As a result, Juárez served as a temporary shelter for more than 19,000 Central American and Mexican migrants awaiting decisions on their U.S. asylum applications [24,25,26]. The foreign affairs secretary, however, did not fulfill Cabada’s wishes, and U.S. citizens continued to pass into Mexico. Consequently, hospitals saw a massive influx of patients. Cemeteries were hosting as many as six burials each day, and the cost to bury a loved one increased by some USD 600 [27]. Unlike in El Paso, people living on the urban periphery of Ciudad Juárez had limited access to clean water and could not practice as effectively the CDC-recommended hygienic practices to counter COVID-19 [21]. They also faced other health barriers, such as the lack of COVID-19 testing sites, which increased the number of cases and deaths in the region [22].

## 3. Data and Methods

The data used in this study were collected through ethno-surveys, which provided close-ended and open-ended data regarding health, employment, socioeconomic status, housing, transborder habits, and citizenship. These surveys were conducted in 2011–2012, nine years before the start of the pandemic, as part of a study funded by the National Institutes for Health (NIH) conducted by Dr. Ernesto Castañeda and research teams who trained for several months as part of a research methods course. The sample consisted of 1152 Hispanic respondents aged 18 and over residing in El Paso, Texas. All surveys were conducted and recorded in Spanish and/or English (at the preference of the respondent) in a variety of locations in El Paso, such as at individual’s homes, shelters, and workplaces. Only close-ended questions from the ethno-surveys are referenced in this study. All researchers who participated in data collection were certified to work with human subjects, and the project had IRB approval from the University of Texas, El Paso. The NIH’s National Institute on Minority Health and Health Disparities further reviewed and approved the project before releasing funds.

The risk assessment comprises several analyses conducted using the IBM SPSS 27 statistical package. Notably, we used purposeful sampling techniques [28] that specifically considered the heterogeneity of respondents, such as education levels, profession, housing status, and age, to construct an adequate representation of El Paso’s geographic neighborhoods. Furthermore, we stopped data collection when saturation of responses was reached and we had a sample large and diverse enough to generalize to the Hispanic population of the whole city. No exclusionary criteria were identified when recruiting participants beyond self-identification as Hispanic, Latino, Mexican American, Chicano, Mexican, or of Latin American or Caribbean origin [6]. 

Undocumented Hispanics and those experiencing housing insecurity were over-sampled because of their relative exclusion across census counts and research studies [29]. Therefore, to take these differential selection probabilities into account and adjust to the El Paso demographic, weighted data were utilized to account for this over-sampling in the dataset. Throughout the following sections, we provide descriptive analyses to present various differences in the distributions of citizenship status, social class, and medical insurance coverage, among other factors, across the sample of Hispanic El Pasoans. Pearson’s chi-squared tests determined bivariate associations.

The data precede the COVID pandemic, but when we embarked on this analysis, we wondered whether they could be used as a baseline to calculate population risks before the pandemic struck El Paso. It is safe to assume that most of the people who participated in the survey still reside in El Paso and that the demographic and health profile has not changed much; if anything, it may have worsened as the people in the sample have aged, but this also depends on the health status of the younger generations for whom comparative data do not exist. Health data are seen as a personal attribute, and patients’ data are protected by HIPA. Even if hospitals engage in big data analysis with their anonymized patient data, they do not have access to all types of populations in a city; thus, the importance of databases such as this. 

## 4. Latinos and High-Risk Populations

### 4.1. Race and Ethnicity

Risk is not only related to physiological factors but also socioeconomic, cultural, racial, and ethnic ones. The CDC distinguishes racial and ethnic minorities, as well as people who are currently homeless, pregnant, and breastfeeding, as individuals who are “high-risk” for COVID-19 [5]. This is because Latin communities across the U.S. have suffered disproportionately high rates of hospitalization and death caused by the virus compared to other racial and ethnic groups. According to the City of New York and the CDC, Black and Latin communities had borne the brunt of virus-related deaths [14]. In a weekly report, the CDC revealed that although 18% of the U.S. population is Black, 33% of hospitalized COVID-19 cases were among Black people in May 2020 [14].

The Latin population in the United States has received relatively little attention in relation to the pandemic despite its vulnerability to COVID-19. This is especially true in the city of El Paso, Texas, where Hispanics constitute over 83% of the population and have suffered disproportionately from the pandemic’s damage [16]. On 28 May 2020, the city recorded 1029 cases and 22 deaths [18]. Over a year later, the cumulative number of infections surpassed 111,000 and the death toll across El Paso had skyrocketed to 1651 lives lost. Hispanic individuals comprised the vast majority of COVID-related deaths at this time, at 91.25% of the population [30].

### 4.2. Essential and Frontline Workers

The Hispanic community is an integral part of the U.S. labor force. In 2018, 17% of the national labor force was comprised of documented Hispanic citizens, 61% of whom were Mexican [31]. However, this percentage does not include undocumented workers, who constitute upwards of an estimated 5.1% of the U.S. workforce [32]. This is supported by our data gathered in El Paso, as reported in Table 1, which show that 53.5% of undocumented Hispanics in 2011 were employed. The way the data were obtained means that the unemployed may be students, homemakers, or retired.

Furthermore, the CDC reports that at least 25% of the Hispanic population in the U.S. is employed in the service industry, including hospitality, transportation/travel, delivery, food, healthcare, and education services [33]. Unfortunately, these industries severely struggled under COVID-19 restrictions and regulations, and nonessential businesses laid off millions of workers across the country as a result of forced closures [34]. This was particularly worrisome for undocumented immigrants because they could not file for unemployment insurance and did not qualify for Pandemic Unemployment Assistance (PUA), which was passed in the Coronavirus Aid, Relief, and Economic Security (CARES) Act [35].

Then, there are those who were considered “essential” employees, including people working in grocery stores, mail services, agriculture, city maintenance, and construction, in addition to the service industries mentioned earlier, who were legally required to work in person during the pandemic. Laws surrounding “essential and emergency employees” require that such employees continue working through national emergencies, including those who are immunocompromised or have pre-existing health conditions [36]. This makes it difficult for frontline workers to receive unemployment benefits if they prefer not to work because one must be fired or laid off in order to collect unemployment insurance [37]. Employees who quit during the pandemic were disqualified from unemployment claims, a policy that jeopardizes their financial stability for the sake of maintaining their health. Therefore, “essential” workers were forced to engage with customers and coworkers in close quarters, as before COVID-19 regulations, and thus ran a higher risk of contracting and spreading the virus. 

### 4.3. Prison, Jail, and Juvenile Delinquent Centers

Incarcerated populations in prisons, jails, and juvenile delinquent centers are also at higher risk of severe illness related to COVID-19. The close quarters, shared spaces, and lack of comprehensive medical care in correctional institutions create a breeding ground for the virus. The Marshall Project reports that there were at least 9437 cases of positive COVID-19 diagnoses in state and federal prisons across the U.S. as of 25 April 2020 [38]. As a result, 131 incarcerated individuals and seven prison employees died [38]. The deaths of prisoners rapidly increased throughout the spring months, eventually amounting to 496 dead inmates by 4 June 2020 [39]. There was no testing protocol for individual states at that time, and many refused to release information regarding the number of prisoners who were tested. The Federal Bureau of Prisons even went so far as to report lower numbers compared to state correctional facilities [40]. 

In the second year of the pandemic, the institution’s reporting habits had not improved. The Federal Bureau released information indicating that 799 federal inmates tested positive for the virus, along with 319 staff members. Although no staff member was among the deceased, 27 inmates died from virus-related complications. Furthermore, three inmates died at Fort Worth Federal Medical Center alone, and all federal prisons in Texas currently had at least one positive diagnosis within the facility [40]. The Marshall Project, however, defines these numbers as “almost certainly an undercount” [38].

This puts racial and ethnic minorities at further risk, given the mass incarceration of communities of color. In the U.S., Hispanics are imprisoned at a rate 1.4 times higher than that of Whites [41]. In 2016, approximately 61% of state prisoners in New Mexico were Hispanic, although approximately 49.1% of people living in New Mexico were Hispanic [42]. The ratio of Black and Hispanic prisoners exceeds that of incarcerated Whites across the country, but the disparities are widest in Texas and other southern states. For example, ranking second only to the number of incarcerated Black individuals, 541 per 100,000 prisoners in Texas are Hispanic [41]. The state’s ratio of Hispanic to White prisoners is 2:1, and its neighbor, Arizona, has the highest number of Hispanic prisoners in the U.S. [41]. 

Juvenile delinquent centers incarcerating children ages 10 to 17 share similar demographic trends with adult correctional institutions. The Department of Justice revealed that Latino youth in the U.S. have a 65% higher chance of being detained and incarcerated than their White counterparts [43]. This profound disparity also exists in Texas, where juvenile prisoners are 1.47 times more likely to be Latino than White [43]. Disproportionate imprisonment because of higher policing and stricter sentencing is also one of the reasons that Black, Hispanic, and other minority communities in the U.S. were contracting, spreading, and dying from COVID-19 at higher rates than White Americans.

### 4.4. Medical Insurance and Citizenship

Immigration status dramatically impacts an individual’s access to healthcare and medical treatment. Our data revealed that the likelihood of being medically insured increased with the stability of immigration status, as detailed in Table 2. In 2011, 89.3% of undocumented Hispanics living in El Paso lacked medical insurance, followed by 66% of lawful permanent residents. 

These data were obtained before the Affordable Care Act was passed. Still, the Affordable Care Act does not provide coverage for all immigrants and excludes the undocumented community from nonemergency services [44]. Recipients of Deferred Action for Childhood Arrivals (DACA) have been denied both Medicaid and ACA benefits since 2012, and children of undocumented parents must be lawful residents or citizens in order to qualify for Medicaid or Children’s Health Insurance Program (CHIP) services [44]. Despite the implementation of the Affordable Care Act, undocumented individuals have not experienced any major improvement in access to healthcare, so it is unlikely that these figures have changed significantly as a result of the Act.

Upon assessment of Hispanics who fulfilled the CDC criteria for individuals at high risk of severe illness related to COVID-19, our data, as shown in Table 3, revealed that only those insured by Medicaid and Medicare were over-represented in the high-risk category. 

### 4.5. Homelessness

The U.S. struggles to provide affordable housing and address homelessness. In 2019, over 500,000 people were estimated to have been homeless on any given night nationwide [45]. People experiencing homelessness are at high risk of severe illness related to COVID-19 for multiple reasons. There are almost no isolated or private spaces in shelters, encampments, and other congregate housing settings that homeless individuals occupy [46]. People are in close proximity to one another, which makes it challenging to maintain the six feet of distance that medical professionals recommend to inhibit the spread of COVID-19. The Council of Economic Advisers found that around 200,000 people sleep in unsheltered places, such as cars, parks, sidewalks, and abandoned buildings. Those who may not sleep in the presence of others must still interact with people at social service facilities [45]. 

People experiencing housing instability are also less likely to have access to masks, gloves, and other materials that prevent the spread of the virus. This also includes basic hygienic facilities and necessities, such as soap [46]. They can also rely on public transportation and facilities, such as bathrooms and water fountains. Interacting without precautionary measures increases the possibility of exposure to COVID-19. According to our data collected in El Paso, housing instability lessens the likelihood of having medical insurance. In 2011, only about one-quarter (24.7%) of homeless Hispanics in El Paso were medically insured, which is a much lower percentage than for those who were housed (see Table 2). 

Additionally, Hispanics experiencing housing instability constituted only 0.3% of those with a high-risk medical condition. However, they are still considered particularly vulnerable because of their heightened likelihood of contracting COVID-19 and limited access to medical providers. Individuals and families experiencing homelessness who are uninsured have a decreased likelihood of being tested for COVID-19 and adequately treated for potentially fatal symptoms (see Table 4). Furthermore, people who have not sought a visit to a medical clinic or provider are unaware of their health status and may be living with an illness that has gone undiagnosed. So, although our data only show 1% of Hispanics experiencing homelessness as “high risk,” the reality is that unhoused people may not know enough about their health to provide information that can assess their level of risk [6]. 

Altogether, experiencing homelessness can almost guarantee poorer health outcomes. Compared to housed individuals, people who have ever experienced homelessness are more likely to face health issues, have unmet healthcare needs, and be subject to accelerated health erosion [47,48,49]. Their obstructed access to basic hygienic amenities, isolated spaces, and medical care makes it challenging for homeless individuals to protect themselves from COVID-19.

### 4.6. Socioeconomic Status

Of the total sample, 52.1% could be classified as low-income and 7.1% as high-income. Racial and ethnic minorities who are low-income are at heightened risk of severe illness related to COVID-19. As shown in Table 5 below, Hispanics with a lower SES are more likely to have a pre-existing high-risk illness that renders them physiologically vulnerable to COVID-19.

It is vital to analyze the risks associated with socioeconomic status, race, ethnicity, and pre-existing health conditions because they are often co-occurring factors that influence an individual or community’s well-being and life expectancy. Similarly, among immigrants, income and employment are intimately associated with citizenship, immigration status, and health insurance. The same is true for people experiencing homelessness, who are typically low-income and vulnerable to COVID-19 [46]. 

## 5. Latinos and High-Risk Pre-Existing Health Conditions

In May 2020, the CDC stated that “older adults and people of any age who have serious underlying medical conditions” are at highest risk of severe illness from COVID-19 [5]. It defines severe illness as “hospitalization, admission to the ICU, intubation or mechanical ventilation, or death” that is caused or exacerbated by the contraction of COVID-19 [50]. Furthermore, the umbrella term of “serious underlying medical conditions” refers to people with chronic lung diseases, severe asthma, heart conditions, obesity, diabetes, immunocompromising illnesses (including, but not limited to chemotherapy/radiation or organ/bone marrow transplantation, HIV/AIDS, or prolonged use of corticosteroids), liver disease, and those undergoing dialysis for chronic kidney disease [5]. The CDC Morbidity and Mortality Weekly Report released on 17 April 2020 disclosed that 89.3% of people hospitalized due to COVID-19 had at least one of the following underlying health conditions: hypertension (49.7%), obesity (48.3%), chronic lung disease (34.6%), type 2 diabetes mellitus (28.3%), or cardiovascular disease (27.8%) [51]. Table 6 shows the distribution of our sample according to our created risk categories.

In the section below, we outline a set of analyses that explore the prevalence of high-risk pre-existing health conditions as they relate to medical insurance status, socioeconomic status, medication, age, and homelessness. We included data on medical insurance, socioeconomic status, and age for all conditions. We included information on medication and homelessness only when the findings were at least statistically significant at *p* < 0.01 or significantly exceeding the average percentage of the sample population. 

Table 7 below reflects the prevalence of high-risk health conditions and diseases in the Hispanic community of El Paso in 2011–2012. 

Two variables were created to paint a clearer picture of Hispanics’ COVID-19 risk profiles. Individuals who had been diagnosed with one or more of the above diseases were categorized into an “at-risk” group, and then further sorted according to their pre-existing condition. Those with two or more, as well as three or more, conditions were grouped into their respective categories because of their compounded physiological vulnerabilities to severe illness related to COVID-19. Emphysema, tuberculosis, and HIV/AIDS were not included in these variables, and because their prevalence is very low, this does not have a significant effect on the results. The table below outlines the percentage of the Hispanic population of El Paso in 2011–2012 that is considered high-risk according to an individual’s number of pre-existing conditions (see Table 8).

Obesity, hypertension, asthma, and type 2 diabetes were the most common illnesses to render Hispanics at high risk. Most notably, however, is that the majority of Hispanic individuals had been diagnosed with one or more health conditions that put them at risk of dying from COVID. Altogether, those with compounded risk comprised about one-third of the total Hispanic population of El Paso at the time that the survey was taken. This indicates poor widespread health outcomes in the community overall and the importance of extreme public health precautions in dealing with respiratory pandemics. 

Additionally, differences in disease prevalence arose when risk was cross-analyzed with place of birth. In Table 9, those with at least one pre-existing health condition were considered higher risk, and those with no pre-existing health condition were categorized as lower risk. Native-born Hispanics made up a slightly smaller portion of high-risk individuals, revealing a lesser likelihood of being diagnosed with a high-risk pre-existing health condition if they had been born in the U.S. These data suggest that birthplace is statistically associated with an individual’s risk of contracting COVID-19. Hispanic immigrants in El Paso showed a collectively higher risk profile.

These disparate health outcomes were more closely examined by separating each specific illness according to individuals’ birthplaces, shown below in Table 10. Hispanics born outside of the U.S. bore the brunt of type 2 diabetes, hypertension, kidney disease, and cancer diagnoses. Foreign-born Hispanics were also slightly more likely to have two high-risk pre-existing health conditions. On the other hand, U.S.-born individuals reported higher rates of asthma and smoking, indicating that immigrant Hispanics and U.S.-born Hispanics struggle with different health concerns in El Paso. 

These findings challenge previous research on what scholars have labeled the “Immigrant Paradox”. A close cousin of the Hispanic Health Paradox, this contradiction arose when researchers began seeing an unexpected pattern in the health outcomes of foreign-born people in the U.S. In general, the Paradox asserts that immigrants exhibit significantly better physical and mental health compared to their native-born counterparts, even within racial and ethnic subgroups of the country’s population [52,53,54,55]. Our data, however, reflect the opposite: Hispanic immigrants were more likely than those born on U.S. soil to be diagnosed with a range and multiplicity of chronic illnesses, all of which are associated with severe sickness upon contraction of COVID-19.

### 5.1. High Blood Pressure

In El Paso, 16.0% of Hispanics reported a high blood pressure diagnosis, and 59.5% of them were low-income. High blood pressure, also known as hypertension, has been coined the “silent killer” given the absence of symptoms that accompany it. Many people go unaware of their high blood pressure, which can lead to the development of kidney disease, cardiovascular diseases, or fatal cardiac events, such as heart attack or stroke [56]. According to our data, 23.2% of adult Hispanics had never had their blood pressure checked. These numbers are concerning because the contraction of COVID-19 is especially dangerous for people with hypertension. The CDC reported that 49.7% of people hospitalized due to virus-related complications had hypertension in March 2020 [51]. Given the high number of those who had never checked their blood pressure, this analysis only accounted for Latin individuals who were aware of their blood pressure status, in addition to a large subsample whose blood pressure was measured as part of the study, as we discuss elsewhere [6].

Although high blood pressure is a cause of serious health conditions, medical coverage varied greatly among those with diagnoses. Our analysis revealed that 33.7% of Hispanics were aware of their high blood pressure but were not medically insured. Even though these individuals knew of their diagnosis, they either chose not to enroll in medical insurance or could not because of cost or citizenship status. 

### 5.2. Cholesterol

High cholesterol was almost equally as common as hypertension within our sample, as 12.9% of Hispanics reported being diagnosed by a health professional. Similar to high blood pressure, people are often unaware of their cholesterol levels until a serious or fatal event occurs. High cholesterol is a predicting factor of heart disease, hypertension, and type 2 diabetes [57]. Low-lipid cholesterols increase plaque growth in the arteries that flow to the brain and heart, eventually accumulating to the point where blood struggles to pass [58]. Heart attack (myocardial infarction) and stroke occur when plaque buildup has completely obstructed the blood from flowing through the arteries [57]. Both are extremely dangerous health conditions that exhibit little to no symptoms, and the lack of knowledge regarding residents’ blood pressure and cholesterol status might render them especially susceptible to health complications caused by COVID-19.

This analysis showed that 28.8% of those diagnosed with high cholesterol did not have medical insurance at the time. More than half (52.8%) of those who had been diagnosed were low-income, whereas 10.2% were high-income. Because of these disparate figures, it is possible that socioeconomic status plays a role in the stress and diet, both of which impact cholesterol levels of Hispanic El Pasoans. Age was also an important factor in analyzing cholesterol, as the majority of cases occurred among people between the ages of 46 and 65. 

### 5.3. Asthma

The study revealed that 7.6% of Hispanics reported asthma diagnoses. When treated and closely monitored, asthma is not life-threatening. However, if someone with asthma has an asthma attack and lacks access to an inhaler or ventilator, then it can be fatal. An attack is caused by severe inflammation that constricts and narrows the air passages that lead to the lungs [59]. Communicable diseases, such as the flu or an upper respiratory infection, can trigger an asthma attack [59]. It is dangerous to be unaware of the condition because those with asthma are at higher risk of complications or death after contracting a communicable illness. Although it is unknown whether COVID-19 induces asthma attacks, shortness of breath and dry cough are common symptoms of the virus that alter the flow of breath through the airways [60]. Severe symptoms and difficulty breathing might trigger an asthma attack, so medical professionals warn individuals with asthma to take caution. Nonetheless, there has been no information that distinguishes asthma attacks from common symptoms of COVID-19. Because of this, virus-related symptoms may be mistaken as a routine asthma attack and deter individuals with asthma from seeking medical attention.

Out of this portion of the Hispanic population in El Paso, approximately half (51.7%) did not have medical insurance to help them manage their asthma diagnosis prior to the Affordable Care Act. An individual’s socioeconomic status also informed the likelihood of receiving an asthma diagnosis, given that 56.3% of Hispanic residents with asthma were low-income and 42.1% middle-income. 

### 5.4. Heart Attack/Stroke

A small percentage (2.6%) of Latin individuals in El Paso reported a previous heart attack or stroke. Given the significance of socioeconomic status and medical insurance in the previous analyses, it is perhaps unsurprising that 27.4% were not medically insured, and an overwhelming percentage (72.8%) were low-income. Socioeconomic status was also associated with the use of medication to treat heart attack or stroke, as half (50.2%) of low-income Latin individuals who received this diagnosis reported being on medication compared to 83.2% of middle-class Hispanics. 

When examining the age groups most affected by these conditions, our data showed that 27.2% of heart attacks or strokes were reported by people between the ages of 60 and 80. Deaths caused by COVID-19 are highest among those over the age of 64, so Latin individuals in this age group who have suffered a heart attack or stroke have compounded risk factors that render them disproportionately vulnerable to COVID-19-related complications [61].

Heart attack and stroke are often caused by high cholesterol, as the low-lipid cholesterols create buildups of plaque that block proper blood flow to the heart and brain [57]. A heart attack is defined as a form of cardiovascular disease by itself, but it may also be an indicator of heart diseases such as arteriosclerosis, diabetes, or coronary artery disease [57]. Further, the CDC reported that 27.8% of COVID-related hospitalizations were among people with cardiovascular disease [51]. Heart attack and stroke are both potentially fatal health events that affect an individual for the rest of their life. For example, Cione’s father experienced three heart attacks before receiving a diagnosis of arteriosclerosis. He was prescribed multiple pharmaceuticals that altered his metabolism, sleep quality, mood, energy levels, and ability to eat. His diet changed drastically, and he must regulate his consumption of cholesterol-rich foods for the remainder of his life. 

The physiological processes leading to stroke are similar to those that result in heart attacks, although the life-long impacts may differ greatly. Depending on the part of the brain that was depleted of blood, stroke can cause paralysis, memory loss, changes in behavior, speech issues, and/or vision impairment. In extreme cases, individuals lose entirely the ability to speak or move their body [62]. Those who are medically uninsured and have experienced one or both of these health events are at higher risk of other health complications, such as an additional heart attack or stroke, if they do not receive proper follow-up care. 

### 5.5. Emphysema

A minority (1.3%) of survey participants had received an emphysema diagnosis. Emphysema is a chronic lung disease that heightens one’s susceptibility to severe illness [51]. Smoking tobacco is the most common cause of emphysema, but air pollution and respiratory infections can also cause or aggravate it. It is defined as a chronic obstructive pulmonary disease (COPD), and people can live with emphysema for years before symptoms develop [63]. Medical treatment typically involves medications, surgery, and oxygen therapy, but it is typical for those with emphysema to forgo these costly treatments [63,64].

Our data revealed that about one-quarter (27.4%) of those who were diagnosed with emphysema was not medically insured. Furthermore, emphysema was associated with lower socioeconomic status, as 54.5% of those with emphysema were low-income, whereas none were high-income. Regarding age, emphysema most commonly affected people ages 60–85, an age group that has been classified as particularly susceptible to severe illness or death related to COVID-19 [5].

### 5.6. Hepatitis or Cirrhosis

A total of 2.2% of the Hispanic population of El Paso reported a diagnosis of hepatitis, cirrhosis, or both. All hepatitis infections (A, B, C, D, and E) are inflammatory and occur in the liver, as well as cirrhosis, as cirrhosis is technically the progression of any liver disease [65]. Hepatitis B and C are the most common causes of cirrhosis, and those who are most at risk of contracting B, C, and D are injection drug users and those who practice unsafe sex [65]. While Hepatitis A is curable, all its other types are not [65]. Injection drug use and unsafe sex are risk factors also associated with the contraction of HIV, and any HIV-positive person who contracts hepatitis is at severe risk of health complications [66]. 

Of those who received a diagnosis, 33.6% are not medically insured. This diagnosis was distributed unequally across socioeconomic classes, given that low-income Hispanics once again constituted the majority at 77.8%, which is also the highest percentage across the illnesses in this risk assessment. Clearly, hepatitis and cirrhosis diagnoses are impacted by an individual’s financial and social standing. This diagnosis also happened to almost exclusively strike young people between the ages of 18 and 30, indicating that younger people may be more susceptible to contracting hepatitis or developing cirrhosis.

### 5.7. Kidney Disease

Around 2.3% of Hispanic study participants reported kidney disease, of whom 42.2% did not have medical insurance and therefore struggled to receive regular care. Many of those with kidney disease diagnoses were middle-income (52.8%) and low-income (41.3%). Kidney disease is an illness that does not exhibit symptoms until the occurrence of a potentially fatal event, such as kidney failure. It is intimately linked with heart disease, diabetes, high blood pressure, and certain forms of cardiovascular disease known to cause or evolve into kidney disease. Further, it is a chronic disease, meaning that the kidneys are permanently damaged and cannot properly filter blood. Unless the patient immediately changes their diet and seeks medical treatment, their condition will worsen with time [67]. A share of 36.4% of Hispanics in El Paso with kidney disease are above the age of 60, which adds another layer of risk in the case of a positive COVID-19 diagnosis.

### 5.8. Cancer

Those with cancer diagnoses amounted to 1.4% of the Hispanic population, which is approximately equal to the proportions of kidney disease and heart attack/stroke. The ethno-surveys also revealed that 16.8% of Hispanic residents of El Paso who had cancer were uninsured, one of the lowest rates of lacking medical insurance across this report. This could be because the culture surrounding cancer in the United States is serious and fearful, which encourages people to remain insured after a diagnosis. People who are nearing remission, are in remission, or may have been diagnosed as a child are also among those who are likely to be insured. 

Approximately half of the Hispanic residents diagnosed with cancer at some point in the past are of low socioeconomic status (50.1%), whereas 40.9% are considered middle-class. However, the ages of people who reported cancer diagnoses at one point in their lives varied greatly: 8.3% were ages 18–25, 16.6% were ages 31–35, and 8.3% were ages 46–50. Ultimately, the highest rates were reported by people between the ages of 51 and 65, who constituted 58.2% of the diagnosed population. This is significant because people ages 65 and older are considered “high risk” by the CDC for COVID-19. Hispanics who also have cancer are more likely to be negatively impacted by COVID-19. Nonetheless, this overall assessment shows how uncommon it is for older Hispanic people in El Paso to have cancer.

### 5.9. HIV/AIDS

The unweighted sample with over-representation of people experiencing homelessness has a 0.2% HIV rate. In our survey, 100% of those with HIV/AIDS were homeless at the time of the survey. Given the high-risk status associated with being homeless and having a positive HIV/AIDS diagnosis, respectively, this portion of the Latin population is extremely vulnerable to health complications or death related to COVID-19.

The percentage of those with HIV/AIDS in our weighted data constitutes about 0.00002% of all Hispanics living in El Paso, which is significantly lower than the 2019 national percentage of 0.34% [66,68]. However, the stigma surrounding HIV/AIDS makes people wary of getting tested and learning about prevention methods. It is estimated that 1 in 7 people living with HIV/AIDS in the United States are unaware of their positive status [66]. Therefore, it is likely that there are other HIV/AIDS-positive Latin people living in the region. 

None of the HIV/AIDS-positive Hispanic residents in the sample were medically insured. Although the number of HIV/AIDS-positive people in the data is small, it is nonetheless worrisome, considering that HIV/AIDS killed over 37,000 people in the U.S. in 2018 [69]. The year before, 53% of new known HIV cases were diagnosed in the South, 21% of which were among Hispanics/Latinos. Although numbers have gradually decreased over the past few years, the rate of new cases in Texas was 15.4 per 100,000 in 2019 [69]. Care for the HIV/AIDS-positive community in the region is also subpar. Recently, the U.S. South reported the lowest number of HIV-positive people who received medical care and had a suppressed viral load from being treated with antiretroviral therapy [69]. Similarly, according to our data, only half of the Hispanic residents with HIV/AIDS were on medication in 2011. Whether the medication being taken was antiretroviral therapy is unknown, so it is possible that even fewer were being treated for HIV/AIDS.

### 5.10. Tuberculosis

The rates of tuberculosis are falling 2% each year globally, but it is still one of the top 10 leading causes of death worldwide [70]. A quarter of the global population has the tuberculosis bacteria lying dormant in their system, so others in the Hispanic community in El Paso may have contracted the bacteria as well. Only 5–15% of these people are estimated to fall ill with tuberculosis, but those with compromised immune systems and pre-existing conditions are at high risk of developing the illness. For instance, HIV/AIDS-positive people are 19 times more likely to die from tuberculosis, which causes further concern that many HIV/AIDS-positive people lack medical insurance [70].

The prevalence of tuberculosis is low in the El Paso community. Only 0.2% of the Hispanic community of El Paso reported having tuberculosis at one point in their lives, approximately half (49.6%) of whom have medical insurance. This is concerning because all of the tuberculosis diagnoses occurred among Hispanic people of low socioeconomic status who might not be able to afford the proper treatment or medicines without pharmaceutical or medical insurance coverage. Individuals over the age of 45 recalled no previous bouts of tuberculosis—instead, almost half of the reported diagnoses occurred among people aged 18–25. 

### 5.11. Diabetes

According to our survey data, 7.3% of Hispanics in El Paso reported a diabetes diagnosis, 60.1% and 3.3% of whom were low-income and high-income, respectively. More than half (65.5%) had medical insurance, but 34.5% were not insured. According to the CDC, diabetes puts one at considerably higher risk of severe health complications related to COVID-19 given that nearly half (49.7%) of people hospitalized with severe virus-related illness as of 30 March 2020 had a previous diabetes diagnosis [51].

Low insulin adherence, which puts individuals in danger of unregulated and dangerous blood sugar levels, can be attributed to multiple factors such as access, affordability, willingness, and guidance on self-administered injection. This was a concern with our sample because only 31.2% of Hispanics with diabetes took insulin at least once. Positive diabetes diagnoses varied considerably according to age, and Hispanics over the age of 60 constituted 29.3% of total cases. This presents evidence that an insulin adherence barrier affected multiple different age groups. Previous research has drawn connections between low insulin usage and depression, embarrassment, a busy schedule, and travel among people living with type 2 diabetes [71,72]. Patients have also reported fears associated with insulin-related weight gain and accidental hypoglycemia, or low blood sugar, induced by an overproduction or excessive dosage of insulin [73]. In a study conducted by Hu et al., Hispanic immigrants with type 2 diabetes conflated insulin therapy with a death sentence, calling it a “last resort” and expressing that they feared injections because the insulin itself might cause further damage [74]. Many participants, especially women, cited a lack of positive family support and access to syringes as barriers to proper insulin use [74].

The survey data draw awareness to a persistent problem affecting the Hispanic community’s well-being. Individuals with diabetes who are not receiving proper insulin treatment, if receiving any at all, will likely be hit harder by a positive COVID-19 diagnosis than those who follow regular schedules and guidelines outlined by a health provider. These are significant findings that reveal the disproportionately poor health experienced by Hispanic individuals living with chronic diseases and inform us of who might be especially vulnerable to COVID-19. 

### 5.12. Obesity

Just over one-quarter (26.2%) of survey participants were considered obese, of whom 3.4% were considered severely obese with a Body Mass Index (BMI) of 40 or higher [51]. Almost half of obese Hispanics lived without medical coverage, and 58.2% of those diagnosed as obese were low-income. Over one-third (38.3%) received a diagnosis between the ages of 18 and 30, although 10.8% of obese Hispanics were ages 61 and older.

Obesity, characterized by a body mass index of 30 or higher, increases a person’s vulnerability to severe illness related to COVID-19 [51]. The CDC reported that 48.3% of individuals hospitalized for virus-related health complications were obese in March 2020 [51]. Before COVID-19, obesity was regarded as a public health issue for the U.S. Hispanic population. In 2019, an estimated 80.4% of Hispanics living in the United States were overweight or obese and were more likely to be obese than White adults [75,76]. These trends are concerning because obesity is associated with many health conditions, including type 2 diabetes, hypertension, stroke, coronary heart disease, sleep apnea, certain cancers, and gallbladder disease [77]. Some of these conditions, as previously discussed, increase the risk of complications from COVID-19.

## 6. Conclusions

Throughout this paper, we dissect the socioeconomic factors that tie into health and well-being and their detrimental effect on racial and ethnic minorities. The COVID-19 pandemic, albeit unexpected, further exposed and reproduced health disparities that were previously less discernible to the general public. The Hispanic community across the United States is already at higher risk of COVID-19 because of institutional discrimination across the sectors of employment, housing, and health. In El Paso, where more than half of Hispanics were of low socioeconomic status, and 48% lacked medical insurance a decade before, their chances of suffering from severe illness related to COVID-19 are even higher. This is particularly dangerous for those with pre-existing health conditions, such as type 2 diabetes, cardiovascular disease, HIV/AIDS, and cancer. It remains unclear whether more Hispanics living in the U.S. will die from COVID-19 than other racial and ethnic groups, as the pandemic has not run its full course. However, preventive measures must be taken in order to protect the Hispanic community in El Paso from tragedy, including the proper allocation of health resources and financial support for low-income, homeless, undocumented, and medically uninsured individuals.

Although discussion of racial disparities is critical, it is just the tip of the iceberg. Health disparities explain how communities of color disproportionately suffer from poor health, but not why state and federal institutions do not properly allocate health resources. The heart of the problem lies in systemic racism, discrimination, and state-sanctioned violence against minorities. Given that communities of color face many structural inequalities, such as poverty, residential segregation, racism, and access to healthcare, they are not to blame for pre-existing or virus-related health disparities [78]. We should take caution in our reporting of racial and ethnic inequities to ensure that data are contextualized within a critical understanding of structural factors that cause disproportionate COVID-19 rates among minority groups. 

In this study, we have outlined how it is not a coincidence that infection and death rates of COVID-19 among Native American, Black, and Hispanic populations in the U.S. have been among the highest in the world since the beginning of the pandemic. Structural inequalities incurred by institutional racism have created, and continue to create, underlying medical conditions and enable increased exposure to the virus, which puts Native American, Black, and Hispanic citizens in far more vulnerable positions regarding COVID-19 than their non-Hispanic White counterparts. When assessing preventative and recovery measures, policymakers and public health officials should consider pre-existing health disparities and their heightened likelihood of working essential or frontline jobs [31,32]. Cities and towns with higher numbers of working-class Latin people should be prepared to conduct extensive community-based health education and outreach through *promotoras de salud* and provide referrals to critical medical care for sub-populations at higher risk. In addition, further research should investigate the lasting effects of COVID-19 infection and “long-COVID” on these groups.

The dataset used to assess the risk that the population of a city had to COVID was collected much before the pandemic started. Nonetheless, this paper shows how similarly detailed data about a city or population can and should be used by public health officials in a preventive fashion to reduce deaths. Public health data should not be limited to prevalence rates, and epidemiology, but should also include social variables and cultural and sociological insights to include the role of beliefs and ideas in health-seeking behavior and the role of socioeconomic factors in producing different health outcomes between and among ethnic and racial groups. Migration and immigration status are essential determinants of health and well-being. Longitudinal health and social data of large and diverse samples that oversample minority, immigrant, and unhoused individuals are an important tool in creating a healthy population.

## Figures and Tables

**Table 1 ijerph-20-00975-t001:** Employment by Citizenship Status among Hispanics in El Paso, TX in 2011–2012.

Status	Citizen	Resident	Undocumented	Visa	Total
Unemployed	33.6%	46.5%	46.5%	19.2%	35.6%
	138,083	36,477	8232	2530	185,322
Employed	66.4%	53.5%	53.5%	80.8%	64.4%
	273,471	41,990	9461	10,655	335,577

Note: *p* < 0.001.

**Table 2 ijerph-20-00975-t002:** Medical Insurance by Citizenship among Hispanics in El Paso, TX in 2011–2012.

Status	Citizen	Resident	Undocumented	Visa	Total
Does not have medical insurance	43.1%	66.0%	89.3%	40.2%	48.0%
	177,533	52,188	15,792	5046	250,559
Has medical insurance	56.9%	34%	10.7%	59.8%	52%
	234,673	26,914	1900	7515	271,002

Note: *p* < 0.001.

**Table 3 ijerph-20-00975-t003:** Health Insurance Plan and High Risk of COVID-19 Complications.

Health Insurance Type	Not at Risk for COVID	At Risk for COVID	Total
Private healthcare plan	51.9%	43.6%	47.7%
	70,011	60,104	130,115
Medicaid	8.3%	12.3%	10.4%
	11,254	16,991	28,245
Medicare	2.3%	9.5%	6.0%
	3125	13,134	16,259
VA	0.9%	2.3%	1.6%
	1250	3135	4385
Tricare	6.0%	3.6%	4.8%
	8125	5010	13,135
Unspecified healthcare plan	28.7%	26.8%	27.8%
	38,758	36,985	75,743
Government employee healthcare	0.9%	0.5%	0.7%
	1250	625	1875
Foreign insurance	0.9%	1.4%	1.1%
	1250	1875	3125

Note: *p* < 0.001.

**Table 4 ijerph-20-00975-t004:** Disease Prevalence among Hispanics Experiencing Homelessness.

Health Condition	Percent	*N*
Smoking	22.5%	469
Obesity	18.5%	438
High Blood Pressure	8.1%	219
High Cholesterol	4.3%	117
Diabetes	4.0%	107
Asthma	3.4%	92
Hepatitis or Cirrhosis	2.8%	76
Heart Attack/Stroke	1.9%	51
Kidney Disease	1.5%	31
Cancer	0.5%	10
Tuberculosis	0.5%	10
Emphysema	0.4%	10
HIV/AIDS	0.4%	10
		1640

Note: Unweighted data was used to include all of the oversampled unhoused respondents.

**Table 5 ijerph-20-00975-t005:** Hispanics with High-Risk Pre-Existing Health Conditions by Socioeconomic Status.

Socioeconomic Status	High-Risk
Low SES	53.8%
	146,205
Medium SES	45.0%
	95,793
High SES	37.4%
	13,790

Note: *p* < 0.001.

**Table 6 ijerph-20-00975-t006:** Number of Pre-Existing Health Conditions Associated with Severe Illness Related to COVID-19 among Hispanics in El Paso in 2011–2012.

Risk Assessment	Percent	*N*
No risk factors	50.70%	266,383
1 risk factor	31.90%	167,396
2 risk factors	15.50%	81,278
3 or more risk factors	2.00%	10,264
		525,321
Not at-risk	50.70%	266,383
At-risk	49.30%	258,938
		525,321
Deaths (as of 22 April 2021)		2506
El Paso County pop. 2019		839,238
Death rate	0.3% of the population	
U.S. Death rate	0.00%	
Global Death rate	~0.4%	

Sources: Reuters Graphics [12,13], U.S. Census Bureau [16].

**Table 7 ijerph-20-00975-t007:** High-Risk Illness and Disease Prevalence among Hispanics in El Paso.

Health Condition	Percent	*N*
Obesity	26.2%	95,818
Smoking	17.7%	92,989
High Blood Pressure	16.0%	83,980
High Cholesterol	12.9%	67,629
Asthma	7.6%	40,100
Diabetes	7.3%	38,235
Heart Attack/Stroke	2.6%	13,810
Kidney Disease	2.3%	11,905
Hepatitis or Cirrhosis	2.2%	11,326
Cancer	1.4%	7510
Emphysema	1.3%	6885
Homelessness	0.5%	2707
Tuberculosis	0.2%	1260
HIV/AIDS	<0.1%	10

**Table 8 ijerph-20-00975-t008:** High-Risk Pre-Existing Health Conditions Associated with Severe Illness Related to COVID-19 Among Hispanics.

Health Condition	Percent	*N*
Obesity (only)	17.4%	45,000
High Blood Pressure (only)	11.3%	29,389
Diabetes (only)	3.9%	9999
Asthma (only)	7.2%	18,754
Heart Attack/Stroke (only)	0.7%	1875
Kidney Disease (only)	0.7%	1875
Cancer (only)	0.5%	1250
Smoking (only)	21.3%	55,043
Homelessness (only)	1.6%	4203
2 of the above	31.4%	81,278
3 or more	4.0%	10,264

**Table 9 ijerph-20-00975-t009:** Hispanics Diagnosed with High-Risk Pre-Existing Health Conditions by Birthplace.

At Risk for COVID-19	Born in the U.S.	Foreign Born	Total
Higher risk	47.4%	52.0%	50.7%
	146,417	112,521	266,383
Lower risk	52.6%	48.0%	49.3%
	162,588	103,795	258,893

Note: *p* < 0.001.

**Table 10 ijerph-20-00975-t010:** Pre-Existing Health Conditions Associated with High Risk of Severe Illness Related to COVID-19, According to Birthplace.

Health Condition	Born in the U.S.	Foreign-Born
Obesity (only)	18.4%	16.1%
High Blood Pressure (only)	11.1%	11.7%
Diabetes (only)	2.1%	6.1%
Asthma (only)	9.4%	4.4%
Heart Attack/Stroke (only)	0.4%	1.1%
Kidney Disease (only)	0.0%	1.7%
Cancer (only)	0.0%	1.1%
Smoking (only)	22.6%	19.5%
Homelessness (only)	1.0%	2.4%
2 of the above	30.5%	32.5%
3 or more	4.4%	3.4%

Note: *p* < 0.001.

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
