# Peer review of "The Implications of Health Disparities: A COVID-19 Risk Assessment of the Hispanic Community in El Paso"

_ijerph, 2023, doi:10.3390/ijerph20020975_

Round 1

Reviewer 1 Report

Overall, this is a very well written case study.

My first major concern is the introduction.  While I agree with the points, I believe the introduction over-focuses on the presidential/government response. These details could serve as a separate background section, allowing the introduction to set the stage for the focus of the study. 

The first paragraph of Section 2, "Latinos and COVID", seemed to provide some of the background information I'd expect to see in the article introduction.  It would be helpful to provide background on the city profiled in the article. It would also help to introduce the "sister city" mentioned later in the article for context. 

There should be more detailed information about the surveys used for the article, such as when and how the surveys were conducted, and if supplementary data sources were used.

The tables are the next concern. In my version of the manuscript, the tables seemed misaligned; I have written a separate note to the editor to determine if this is how the draft was submitted, or simply due to the way review drafts are presented.

Aside from formatting, the tables in the article should be tied into the text a little better. Most tables are not referenced by the text, and some could possibly be removed. 

Tables 2-4 focus on another city, New York City, and could possibly be removed.  It is fine to keep the mention of the statistics as an example, but the table content is less relevant to the body of the article.  In addition, the accompanying text seems to have been written in June 2020 (as illustrated by the sentence: "Just two months later, the mortality and infection rates have skyrocketed").  As it is now over 2 years later, the tense and content of this and other sections should be updated. 

Some of the other tables are not directly referenced in the text.  This is particularly an issue when it is unclear, as in Tables 7,8,10, and 14, whether the data is specific to El Paso or nationwide. There are also no dates mentioned in the captions of those tables. Since many of the other tables are sourced from 2010-2011 data, it would be helpful to add dates/time periods to all table captions for context. 

Finally, the data used for the article all seems to have been collected in 2010-2012, ten years before the pandemic formally began. There could have been significant shifts in the demographic profile of El Paso in that time. Is there more current data available now? If not, this limitation should be acknowledged/addressed.

Author Response

We thank very much the reviewers for the feedback.

As reviewer 1 writes, “Overall, this is a very well written case study.”

“While I agree with the points, I believe the introduction over-focuses on the presidential/government response. These details could serve as a separate background section, allowing the introduction to set the stage for the focus of the study. The first paragraph of Section 2, "Latinos and COVID", seemed to provide some of the background information I'd expect to see in the article introduction.” 

Authors: We have moved the policy response to a new background section and have foregrounded the discussion on health disparities earlier in the introduction.

“It would be helpful to provide background on the city profiled in the article.”

AU: We added a paragraph and a reference “Poverty, Place, and Health along the US-Mexico Border,” for reasons of space we cannot add much more context on El Paso and Ciudad Juarez here, but we will do so elsewhere in a follow-up longer publication.

Some of the text added is: El Paso is city in the western corner of Texas, bordering Las Cruces, New Mexico and directly to the south Ciudad Juarez, Mexico. El Paso City’s population was around 649 thousand in 2010 and 679,000 in 2020 and El Paso County has over 800,000 residents. El Paso is around 83% Hispanic. El Paso is a majority Latin city, with over 24% of the population being foreign-born. It also includes Hispanics who have been in the area for decades and many generations. Thus, making it a great place to study the impact of health disparities within the same ethno-racial category.  

“There should be more detailed information about the surveys used for the article, such as when and how the surveys were conducted, and if supplementary data sources were used.”

AU: You are absolutely right. We added a methods section.

“In my version of the manuscript, the tables seemed misaligned; I have written a separate note to the editor to determine if this is how the draft was submitted, or simply due to the way review drafts are presented.”

AU: We have worked on the format of the tables, and the editors may do further page setting.

“The tables in the article should be tied into the text a little better. Most tables are not referenced by the text, and some could possibly be removed. Tables 2-4 focus on another city, New York City, and could possibly be removed.  It is fine to keep the mention of the statistics as an example, but the table content is less relevant to the body of the article.”

AU: We have simplified many tables and removed some tables including the one with data from New York.

“As it is now over 2 years later, the tense and content of this and other sections should be updated.”

AU: We have changed the tense to the past throughout the paper. 

“Some of the other tables are not directly referenced in the text.  This is particularly an issue when it is unclear, as in Tables 7,8,10, and 14, whether the data is specific to El Paso or nationwide. There are also no dates mentioned in the captions of those tables. Since many of the other tables are sourced from 2010-2011 data, it would be helpful to add dates/time periods to all table captions for context.”

AU: All tables refer to data from El Paso collected during the 2010-2011 period.

“Finally, the data used for the article all seems to have been collected in 2010-2012, ten years before the pandemic formally began. There could have been significant shifts in the demographic profile of El Paso in that time. Is there more current data available now? If not, this limitation should be acknowledged/addressed.

AU: The data precedes the COVID pandemic, but it served as a useful baseline to talk about risks before the pandemic struck El Paso. It is safe to assume that most of the people who participated in the survey still reside in El Paso and that the demographic and health profile has not changed much, if anything it may have worsened as the people in the sample have aged, but this also depends on the health status of the younger generations for whom comparative data does not exist. Health data are seen as a personal attribute, and patients’ data is protected by HIPA. Even if hospitals engage in big data analysis with their anonymized patient data, they do not have access to all types of populations in a city. Thus, the importance of databases such as this.

We added to the conclusion:

The dataset used to assess the risk that the population of a city had to COVID was collected much before the pandemic started. Nonetheless, this paper shows how similarly detailed data about a city or population can and should be used by public health officials in a preventive fashion to reduce deaths. Public health data should not be limited to prevalence rates, and epidemiology, but also include social variables and cultural and sociological insights to include the role of beliefs and ideas in health-seeking behavior and the role of socioeconomic factors in producing different health outcomes between and among ethnic and racial groups. Migration and immigration status are essential determinants of health and well-being. Longitudinal health and social data of large and diverse samples that oversamples minority, immigrant, and unhoused individuals are an important tool in creating a healthy population.

Thanks again for your pointed and correct observations and questions.

Reviewer 2 Report

I am glad for revising this manuscript. Despite lacking clarification regard study design, which I strongly recommend adding for a better understanding,  the paper provided a relevant discussion about role of inequities on health status, especially during COVID-19 pandemic.

I pointed out below a few suggestions for improving this manuscript. 

In introduction section, I suggest that the authors made clear the approach to address this topic. In addition, I also recommend the authors revising the links between these different aspects discussed. For instance, subtitles 2 and 3 could be blended. Conversely, topic 6 would be better placed prior in the manuscript and, hence, help the readers linking the information throughout the whole text.

Some topics seemed lack proper references. Therefore, the authors should carefully revise the manuscript. Similarly, tables present some formatting error since lines are absent.

Topics 2 to 8 could be under topic 1. The conclusion should be summarized mentioning only the main purpose of this text. It also could be interesting to provide some practical implications, including the social and academic impact, and strenghts and potential limitations as well.

Author Response

Reviewer 2 wrote, “I am glad for revising this manuscript…the paper provided a relevant discussion about role of inequities on health status, especially during COVID-19 pandemic.”

AU: Thank you for agreeing to review the paper, for the support, and useful comments.

“I pointed out below a few suggestions for improving this manuscript. In introduction section, I suggest that the authors made clear the approach to address this topic.”

AU: We have changed the order of the introduction, framed the health disparities discussing forefront, added background info on policies and the city, and clarified how the data was used as a baseline exercise that proved correct.

“In addition, I also recommend the authors revising the links between these different aspects discussed. For instance, subtitles 2 and 3 could be blended. Conversely, topic 6 would be better placed prior in the manuscript and, hence, help the readers linking the information throughout the whole text.”

AU: We have changed the order of some of the sections, the place of some of the subtitles and what they contain in order to improve clarity and readability. We moved the discussions of race and ethnicity as well as citizenship earlier on, to further help frame the paper and its findings.

“Tables present some formatting error since lines are absent.”

AU: The table formatting has been improved; the journal editors will solve remaining formatting issues.

“Topics 2 to 8 could be under topic 1.”

AU: The new section 5. titled LATINOS AND HIGH-RISK PRE-EXISTING HEALTH CONDITIONS, includes the results for each condition as a subtopic. Thank you for the helpful suggestion.

“The conclusion should be summarized mentioning only the main purpose of this text. It also could be interesting to provide some practical implications, including the social and academic impact, and strengths and potential limitations as well.”

AU: We added the following phrases,

To the methods section:

The data precedes the COVID pandemic, but when we embarked on this analysis, we wondered whether it could be used as a baseline to calculate population risks before the pandemic struck El Paso. It is safe to assume that most of the people who participated in the survey still reside in El Paso and that the demographic and health profile has not changed much, if anything, it may have worsened as the people in the sample have aged, but this also depends on the health status of the younger generations for whom comparative data does not exist. Health data are seen as a personal attribute, and patients’ data is protected by HIPA. Even if hospitals engage in big data analysis with their anonymized patient data, they do not have access to all types of populations in a city. Thus, the importance of databases such as this.

To the conclusion:

The data used to assess the risk that the population of a city had to COVID was collected much before the pandemic started. Nonetheless, this paper shows how similarly detailed data about a city or population can and should be used by public health officials in a preventive fashion to reduce deaths. Public health data should not be limited to prevalence rates, and epidemiology, but also include social variables and cultural and sociological insights to include the role of beliefs and ideas in health-seeking behavior and the role of socioeconomic factors in producing different health outcomes between and among ethnic and racial groups. Migration and immigration status are essential determinants of health and well-being. Longitudinal health and social data of large and diverse samples that oversamples immigrant and unhoused individuals is an important tool in creating a healthy population.

Thank you for the useful feedback and for signing your review report.

Round 2

Reviewer 1 Report

Thanks for your careful attention to the suggestions. This will be a great contribution to the body of knowledge on this topic!